# Protective Effect of *Lactobacillus crispatus* against Vaginal Colonization with Group B Streptococci in the Third Trimester of Pregnancy

**DOI:** 10.3390/pathogens11090980

**Published:** 2022-08-27

**Authors:** Maja Starc, Miha Lučovnik, Petra Eržen Vrlič, Samo Jeverica

**Affiliations:** 1Department of Obstetrics and Gynecology, General Hospital Slovenj Gradec, 2380 Slovenj Gradec, Slovenia; 2Department of Perinatology, University Medical Centre Ljubljana, 1000 Ljubljana, Slovenia; 3Department of Obstetrics and Gynecology, Community Health Center Ljubljana, 1000 Ljubljana, Slovenia; 4Center for Medical Microbiology, National Laboratory of Health, Environment and Food, 2000 Maribor, Slovenia

**Keywords:** group B Streptococcus, *Lactobacillus crispatus*, colonization, vaginal microbiota, semiquantitative culture, pregnant women, invasive GBS infections, neonates

## Abstract

Background: A normal vaginal microbiota may protect the vaginal mucosa from colonization by potentially pathogenic bacteria, including group B streptococci (GBS). The aim of this study was to investigate the association between colonization with GBS and the presence of specific vaginal microbiota isolated from vaginal swabs in the third trimester of pregnancy. Methods: A semiquantitative culture of 1860 vaginal swabs from consecutive pregnant women in their third trimester was analyzed. The dominant bacteria, including lactobacilli, were identified using MALDI-TOF mass spectrometry. An enrichment culture for GBS was performed on the swabs. GBS colonization correlated with the bacteria isolated at the same time. Results: *Lactobacillus*
*crispatus* was isolated in 27.5% of the cultures, followed by *L. jensenii* (13.9%), *L. gasseri* (12.6%), and *L. iners* (10.1%). The presence of lactobacilli as a group, and of *L. crispatus,* inversely correlated with GBS colonization (OR = 0.44 and OR = 0.5, respectively; both with *p* < 0.001). Other microorganisms, including *Gardnerella vaginalis*, mixed aerobic bacteria and yeasts, were not associated with GBS colonization. Conclusions: Lactobacilli, especially *L. crispatus*, may prevent GBS colonization in pregnancy. Maintaining a normal vaginal microbiota could be an effective method for the antibiotic-free prevention of invasive GBS infections in neonates.

## 1. Introduction

Group B Streptococcus (GBS) is considered an opportunistic type of bacteria that can be part of the normal vaginal and rectal microbiota. Maternal colonization with GBS predisposes to a number of adverse pregnancy outcomes, including stillbirth, preterm birth, and neonatal invasive GBS disease, which may contribute to perinatal mortality or cause long-term neurodevelopmental disorders in the child, and may also lead to invasive disease in the mother [1,2,3,4].

The prevention of GBS infection in pregnancy focuses mainly on reducing the risk of GBS transmission from mother to newborn during labor and delivery with intrapartum antibiotic prophylaxis, based on either clinical risk factors or microbiologic testing. This approach can prevent many cases of invasive disease in newborns, but cannot eliminate all serious infections [5,6,7]. As a result, extensive research has been conducted on the pathogenesis of GBS colonization and the mechanisms of ascending infection, focusing on the interactions between bacterial virulence factors and various aspects of host defenses [8].

The vaginal microbiota may be an important determinant in the host that modulates the colonization rate of GBS. Using a molecular approach, five major types of vaginal microbiota have been described, dominated by the four major vaginal lactobacilli, namely, *Lactobacillus crispatus*, *Lactobacillus jensenii*, *Lactobacillus gasseri*, and *Lactobacillus iners*, while the fifth type is characterized by the absence of lactobacilli and the abundance of mixed anaerobic bacteria, which correspond to microbiota characteristic of bacterial vaginosis [9]. It has also been shown that not only the presence of bacterial species, but also their abundance are important for the classification of vaginal microbiota into these five types [9,10,11]. With the introduction of MALDI-TOF mass spectrometry in clinical bacteriology, the identification of bacteria, including all major types of vaginal lactobacilli, has become accurate and feasible [12]. In addition, semiquantitative cultures by streaking in four quadrants can be used to determine the relative abundance of major microbial species, identify dominant bacteria in the vaginal microbial community, and tentatively classify the type of vaginal microbiota.

Several in vitro studies have attempted to determine the effect of lactobacilli on other members of the vaginal microbiota [13,14,15,16]. The main protective mechanisms of lactobacilli are the prevention of adherence through coaggregation and biofilm formation; production of antibacterial compounds, such as lactic acid, hydrogen peroxide, and bacteriocins; and the stimulation of local immune responses [14]. There are relatively few in vivo studies describing the relationship between GBS colonization and other dominant members of the vaginal microbiota or microbiota types, with largely conflicting results [11,17,18,19]. They used either culture-dependent or culture-independent techniques. Even fewer studies specifically addressed the population of pregnant women [17,19,20].

The aim of this study was to investigate the relationship between GBS colonization and the presence of specific vaginal microbes isolated from vaginal swabs in the third trimester of pregnancy.

## 2. Materials and Methods

### 2.1. Study Design and Population

This was a retrospective cross-sectional analysis of data based on consecutive microbiological cultures in pregnant women in the third trimester between 28 and 42 weeks of gestation. Microbiological results of vaginal swabs from two obstetrics and gynecology centers in Slovenia, namely, The University Medical Centre Ljubljana and Community Health Centre Ljubljana, were reviewed from 2014 to 2018. A high vaginal or cervical swab was taken under aseptic conditions by the physician during prenatal care in a symptomatic or asymptomatic pregnant woman. In the case of multiple swabs, only the first one per pregnancy was included in the analyses. Combined rectal and vaginal swabs taken for GBS screening were excluded from the analyses.

### 2.2. Microbiological Methods

Vaginal swabs in Amies transport medium were used for the study and transported to the laboratory within 24 h of collection at room temperature. Semiquantitative relative cultures were performed with inoculation in 4 quadrants. Swabs were inoculated onto 5 agar media: blood agar (BA), chocolate agar (CA), colistin–nalidixic acid agar (CNA), ChromID Strepto B (STRB; bioMérieux, Marcy l’Etoile, France), and Gardnerella vaginalis Selective Agar supplemented with human blood (GARD; Oxoid, Basingstoke, UK). GBS colonization was determined from vaginal swabs with an enrichment culture. Briefly, Todd-Hewitt broth supplemented with 15 µg/mL nalidixic acid and 10 µg/mL colistin (THBS; Oxoid, Basingstoke, UK) was inoculated, and 10 µL of the overnight culture was transferred to another ChromID Strepto B agar for an additional 2 days. Plates were incubated in normal atmosphere (BA, CNA, STRB) and in 5% CO_2_-enriched atmosphere (CA, GARD) and examined for growth after 24 and 48 h. All typical colonies were identified using the MALDI Biotyper (Bruker Daltonics, Hamburg, Germany). The extent of growth was determined separately for each species as 1+, 2+, or 3+ when bacteria grew in the first, second, and third/fourth quadrants, respectively. When GBS was present after the enrichment procedure, this was recorded as 1+.

### 2.3. Data Analysis

A Microsoft Excel spreadsheet was generated containing anonymized demographic data and microbiological data. Descriptive statistics, prevalence estimations, and group comparisons were calculated using JASP 0.16 open source statistical package for the statistical analysis. A Chi-squared test was used for the comparison of proportions among independent groups. The logistic regression model and crude odds ratio were calculated for binominal data.

## 3. Results

### 3.1. Demographic Data and Microbiological Characteristics

Demographic characteristics are shown in Table 1. Briefly, we examined 1860 consecutive vaginal swabs from pregnant women in the third trimester of pregnancy. The mean age of pregnant women at the time of swab collection was 31 years (17–47 years). The mean gestational age at the time of sample collection was 33 weeks (28–41 weeks), and the median was 33 weeks. GBS colonization was not associated with age or gestational age.

Altogether, 2706 bacterial isolates were recovered in semiquantitative cultures, as shown in Table 1. Lactobacilli were the predominant bacterial species recovered overall, with 1349 isolates. *L. crispatus* was the most common among them (n = 512), followed by *L. jensenii* (n = 259), *L. gasseri* (n = 234), *L. iners* (n = 188), and other lactobacilli as a group (n = 156), most of which were *Lactobacillus delbrueckii* and *Lactobacillus rhamnosus*. Heavy growth of lactobacilli, recorded semiquantitatively as 3+ or 2+ growth, was detected in 82% (n = 1106) and 17.3% (n = 234). Mostly, in more than 99% of lactobacilli-positive cases, they were present as a single lactobacilli species (data not shown).

*Gardnerella vaginalis* was the most common nonlactobacilli bacteria, with 314 isolates recovered. Most of them were detected as heavy growth of 3+ and 2+, with 85.3% (n = 268) and 14.3% (n = 45), respectively. In the mixed aerobic bacteria group, the most common isolate was GBS (10.2%; n = 275); however, an enrichment culture was used to detect GBS, which can also detect lower quantities of bacteria. Of these, 42.5% (n = 117) were recovered in low quantities and recorded as 1+. Finally, 376 yeast isolates were detected, representing 13.9% of all isolates. The microbiological characteristics with the semiquantitative quantification of isolates and groups are shown in Table 1.

More than one species per vaginal culture was recovered in 39.6% (n = 736) of cultures. Of those, five, four, three3, and two isolates were recovered in 0.2% (n = 4), 1% (n = 18), 8.5% (n = 158), and 29.9% (n = 556) of cultures, respectively. One species was recovered in 55.1% (n = 1024) of cultures (data not shown).

### 3.2. GBS Colonization in Different Vaginal Microbiota Environment

The correlation between GBS colonization and the detection of most common vaginal isolates is shown in Table 2. At least one species of lactobacilli was detected in 71.3% (n = 1327) of vaginal cultures. The four major species of vaginal lactobacilli, namely, *L. crispatus*, *L. jensenii*, *L. gasseri*, and *L. iners*, were detected in 27.5%, 13.9%, 12.6%, and 10.1% of cultures, respectively. *Gardnerella vaginalis* was detected in 16.9% of cultures, while the culture positivity rate of mixed aerobic bacteria and yeast was 11.3% and 20.2%, respectively.

Colonization with GBS, as detected with enrichment cultures from vaginal swabs, was significantly dependent on the presence of other members of the vaginal microbiota. Thus, the presence of any lactobacillus or *L. crispatus* inversely correlated with GBS recovery with odd ratios of 0.44 and 0.50, respectively (both with *p* < 0.001). Other lactobacilli, *G. vaginalis*, mixed aerobic bacteria as a group, and yeasts were not significantly correlated with GBS detection. The overall GBS positivity rate in vaginal swabs was 14.8% (n = 275).

## 4. Discussion

The microbes that make up the vaginal environment, particularly lactobacilli, are important in the dynamics of GBS colonization in pregnancy. We reported a lower vaginal colonization with GBS when lactobacilli were the dominant part of the vaginal microbiota, with the strongest negative correlation for *L. crispatus*, arguably the most protective vaginal lactobacillus species. Furthermore, our results confirmed that culture-based methods, especially when used in conjunction with the detailed identification of lactobacilli with MALDI-TOF mass spectrometry and semiquantitative inoculation, can be valuable predictors of the type of vaginal microbiota, the dominant bacteria in the microbial community, and their impact on health and disease.

Maternal vaginal and rectal GBS colonization is one of the best-defined risk factors for fetal and neonatal adverse outcomes [21]. In the perinatal period, 18% of women are carriers of GBS, with regional variations ranging from 11 to 35% [1,22]. The introduction of intrapartum antibiotic prophylaxis, based on universal GBS screening or the presence of risk factors, primarily reduced the burden of early-onset invasive GBS disease in newborns, which can develop in the first 7 days of life [21]. In addition, this strategy relies on the use of antibiotics, and as more pregnant women receive antibiotics, antibiotic resistance develops. Antibiotic use during pregnancy and labor can also affect the health of the newborn as the microbiome changes in early life [5]. Therefore, antibiotic-independent prevention strategies targeting any point of GBS transmission from mother to child or preventing the development of invasive disease in the newborn are needed.

Lactobacilli are involved in maintaining the balance of human vaginal microbiota by preventing the overgrowth of pathogenic and opportunistic microorganisms. They reduce the viability of GBS through various mechanisms, such as the production of lactic acid and lowering the pH of the vaginal environment. They produce hydrogen peroxide and other antibacterial substances, attach to vaginal epithelial cells, and form a protective biofilm, while inhibiting pathogen adhesion, thus, protecting the vaginal epithelial barrier from colonization and invasion by pathogens. Studies have shown that lactobacilli can also exert an immunomodulatory effect in vaginal infection models and activate the local immune system, enhancing the host response to urogenital pathogens [13,14,15,16].

During normal pregnancy, the vaginal microbiota is very stable, has a low bacterial diversity, and is associated with high levels of lactobacilli, especially *L. crispatus* [23,24,25]. In our sample, *L. crispatus* was the most prevalent among the lactobacilli, as almost 30% of all samples were dominated by this species, and it was the only bacterial species that showed a negative association with GBS colonization. This result was expected, since the *L. crispatus*-dominated vaginal microbiota, also known as the community state type I (CST I) [13], has been associated with a healthy vaginal environment in many studies. It is known for its protective effects against dysbiotic vaginal pathogens and sexually transmitted infections [26,27,28]. Other lactobacilli isolated from pregnant women appear to have less, or no, antagonistic effect on GBS colonization. Interestingly, of the four lactobacilli, *L. gasseri* had the least antagonistic effect on GBS colonization, whereas *L. jensenii* and *L. iners* probably had only a marginal effect themselves, as indicated by the trend, which, however, did not reach statistical significance.

With regard to other common vaginal pathogens, some studies describe *Candida* spp. colonization in women as a risk factor for GBS carriage [29,30], but this was not the case in our dataset, as a positive association between GBS and vaginal yeast colonization was observed but was not statistically significant. Animal studies have shown that exposure to *G. vaginalis* promotes vaginal colonization with GBS, leading to a possible increased likelihood of invasive perinatal GBS infection [31]. In contrast, our data showed a nonsignificant negative correlation. This was consistent with some previous studies that also showed a negative or no correlation between bacterial vaginosis and GBS colonization [17,32].

The results of our study were essentially consistent with the relatively few other culture-based studies that have shown an inverse relationship between GBS colonization and lactobacilli [17,18]. Our study included nearly 1900 women in the third trimester of pregnancy and used both semiquantitative cultures, which allowed us to determine the dominant member of the vaginal microbiota present in high abundance (3+ or 2+), and the detailed identification of lactobacilli with MALDI-TOF mass spectrometry, which allowed for species-level correlations and the determination of the distribution of dominant lactobacilli. In combination, this setting provided a robust and reliable estimate of true correlations.

The introduction of culture-independent techniques greatly improved our knowledge of the composition of the vaginal microbiota and the interactions between different members of microbial populations [9]. Currently, there are two major molecular techniques for profiling microbial communities: the amplification and sequencing of the hypervariable region of the 16S rRNA gene and WGS metagenomics, which is widely considered the gold standard for this application [20]. Rosen et al. applied the 16S rRNA method to an ethnically mixed group of nonpregnant premenopausal women (n = 432). Samples were assigned to the five previously defined CSTs, and no association was found between the CST and GBS colonization. They concluded that it is unlikely that lactobacilli play an important role in GBS colonization in vivo [11]. Pace et al. studied a well-defined sample of pregnant women using the WGS metagenomics approach, and showed that this method could profile the microbial community down to the species and even the strain level, and reliably predict the status of vaginal GBS colonization when compared with the enrichment culture. Moreover, a negative correlation between *L. crispatus* and GBS colonization was consistently found, confirming previous culture-based observations, as well as the main findings of our study [20].

There were several limitations in our study. First, this was a culture-based study, in which we selected only the dominant culturable species in the sample and disregarded possible minor components of the vaginal flora that could be detected with a molecular approach. In addition, we only used selected media types that allowed for the detection of previously determined dominant vaginal bacteria, and we did not perform anaerobic cultivation. Nevertheless, we believe that both the semiquantitative approach to cultivation and the MALDI-TOF mass spectrometry identification represent an important advance and improvement in reliability compared to previously published culture-based studies. Second, GBS colonization was determined only with vaginal swabs, meaning that the rectal colonization was not measured. Using this approach, GBS colonization was several points lower than would have been measured if both niches had been sampled. However, the true coincidence could be better assessed with this approach, as the concentration of lactobacilli in the gastrointestinal tract is less well understood. Third, yeasts were not identified as species in our study, so differences in the correlations between *C. albicans*, *C. glabrata*, and possible other yeast species could not be determined. Because previous studies have shown that *C. albicans* is a major causative agent of vulvovaginal candidiasis in pregnancy (in approximately 90% of cases), our results roughly applied to this yeast species as well. Fourth, no additional clinical data were collected because this was a microbiota study. Based on our clinical practice, the sample included a balance of symptomatic and asymptomatic pregnant women and provided a good approximation of the normal pregnant vaginal microbiota.

The prevention of maternal GBS colonization could reduce the need for intrapartum antibiotic prophylaxis. Maintaining an *L. crispatus*-dominant vaginal microbiota or introducing specific *L. crispatus*-derived molecules could be used as a complementary alternative to conventional antibiotics to prevent GBS colonization, mother-to-child transmission, and to further reduce the incidence of severe perinatal infections, preterm delivery, or stillbirth. Our comprehensive clinical data supported the existing evidence on the anti-GBS activity of *L. crispatus* in the vaginal microbiota as a host defense mechanism.

The effectiveness of a vaginal application of *L. crispatus* CTV-05 has already been demonstrated in preventing the recurrence of bacterial vaginosis in nonpregnant women [33]. In addition, some studies have used other Lactobacillus species to reduce vaginal GBS colonization. The probiotic *L. salivarius* CECT-9145 reduced rectovaginal GBS colonization during 26-38 weeks of gestation [34]. *L. rhamnosus* GR-1 and *L. reuteri* RC-14 taken orally during the third trimester reduced the rate of GBS rectovaginal colonization at delivery [35,36]. In mice, serial vaginal inoculation with the probiotic *L. reuteri* CRL-1324 induced partial protection against GBS [37]. In the absence of sufficient data to support the safe and effective use of the oral or vaginal administration of lactobacilli to minimize the risk of GBS colonization, larger studies are needed to provide further evidence of the utility of such an approach [38]. We also need to determine the best composition of lactobacilli, i.e., whether one strain or a selection of different strains—a lactobacilli cocktail—would give the best results. In addition, we need to determine which route of administration of protective lactobacilli would provide the best protection, whether oral or vaginal.

## 5. Conclusions

Vaginal lactobacilli are important bacterial species that limit GBS colonization. We found that pregnant women with high numbers of vaginal lactobacilli, particularly *L. crispatus*, were less likely GBS carriers in the third trimester of pregnancy. Other components of vaginal microbiota, alone or as a group, may be less important. These results suggested that the vaginal microbiota plays an important role in GBS colonization and, possibly, in adverse pregnancy outcomes such as neonatal GBS infections.

## Figures and Tables

**Table 1 pathogens-11-00980-t001:** Demographic and microbiological characteristics.

Parameter	Frequency; n (%)	Quantity of growth; n (%)		
		3+	2+	1+
**Demographic characteristics**				
Swabs	1860 (100)	nd	nd	nd
Patients	1831 (100)	nd	nd	nd
Maternal age (mean)	31	nd	nd	nd
Gestational age (median)	33	nd	nd	nd
**Microbiological isolates (n = 2706)**				
**Lactobacilli**	1349 (49.9)	1106 (82.0)	234 (17.3)	9 (0.7)
Lactobacillus crispatus	512 (18.9)	441 (86.1)	70 (13.7)	1 (0.2)
Lactobacillus jensenii	259 (9.6)	234 (90.3)	24 (9.3)	1 (0.4)
Lactobacillus gasseri	234 (8.6)	188 (80.3)	44 (18.8)	2 (0.8)
Lactobacillus iners	188 (6.9)	128 (68.1)	56 (29.8)	4 (2.1)
Others	156 (5.8)	115 (73.7)	40 (25.6)	1 (0.6)
**Gardnerella vaginalis**	314 (11.6)	268 (85.3)	45 (14.3)	1 (0.3)
**Mixed aerobic bacteria**	667 (24.6)	369 (55.3)	170 (25.5)	128 (25.5)
Streptococcus agalactiae (GBS)	275 (10.2)	94 (34.2)	64 (23.3)	117 (42.5)
Enterococcus faecalis	113 (4.2)	97 (85.8)	16 (14.2)	0 (0)
Streptococcus anginosus	95 (3.5)	61 (64.2)	32 (33.7)	2 (33.7)
Escherichia coli	84 (3.1)	66 (78.6)	18 (21.4)	0 (0)
Staphylococcus aureus	34 (1.3)	13 (38.2)	13 (38.2)	8 (23.5)
Others	66 (2.4)	38 (57.6)	27 (57.6)	1 (1.5)
**Yeast**	376 (13.9)	148 (39.4)	160 (42.5)	68 (18.1)

Note: All isolates are included from 1860 swabs collected. More than 1 isolate per culture could be recovered. nd—not determined.

**Table 2 pathogens-11-00980-t002:** Correlation between GBS and other members of vaginal microbiota in pregnant women.

	Positive Culture	GBS Colonization in Correlation with Culture Result ^#^		OR (95% CI)	*p*-Value
		Positive	Negative		
	n (%)	n (%)	n (%)		
**Lactobacilli any**	1327 (71.3)	153 (11.5)	122 (22.9)	0.44 (0.34–0.57)	< 0.001
Lactobacillus crispatus	512 (27.5)	47 (9.2)	228 (16.9)	0.50 (0.36–0.69)	< 0.001
Lactobacillus jensenii	259 (13.9)	28 (10.8)	247 (15.4)	0.66 (0.44–1.01)	0.054
Lactobacillus gasseri	234 (12.6)	40 (17.1)	235 (14.5)	1.22 (0.85–1.76)	0.288
Lactobacillus iners	188 (10.1)	20 (10.6)	255 (15.3)	0.66 (0.41–1.07)	0.093
Other lactobacilli (group)	146 (7.8)	20 (13.7)	255 (14.9)	0.91 (0.56–1.48)	0.700
**Gardnerella vaginalis**	314 (16.9)	39 (12.4)	236 (15.3)	0.79 (0.55–1.13)	0.196
**Mixed aerobic bacteria ^$^**	211 (11.3)	34 (16.1)	241 (14.6)	1.12 (0.76–1.66)	0.564
**Yeast**	376 (20.2)	66 (17.5)	209 (14.1)	1.30 (0.96–1.76)	0.091

^$^ Mixed aerobic bacteria group included members of *Enterobacterales*, *Enterococcus* spp., and other aerobic bacteria, but excluded GBS and *S. anginosus* group of bacteria. ^#^ GBS colonization in vaginal swabs was 14.8% overall (n = 275). Note that only vaginal swabs were evaluated with the enrichment culture.

## Data Availability

For review of raw data, please contact the corresponding author.

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
