# Peer review of "Protective Effect of Lactobacillus crispatus against Vaginal Colonization with Group B Streptococci in the Third Trimester of Pregnancy"

_pathogens, 2022, doi:10.3390/pathogens11090980_

Round 1

Reviewer 1 Report

Group B streptococci have been known to cause adverse pregnancy outcomes and neonatal disease. The authors have sought to determine the protective effect of Lactobacilli against vaginal colonization with GBS thereby emphasizing an effective method for antibiotic-free prevention of invasive GBS infections in neonates.

While the findings are novel, the authors should refer to STROBE guidelines for reporting observational studies and restructure the manuscript accordingly. May not include all points but some revisions would enhance the quality of this article. The conclusions are otherwise consistent with the evidence presented.

Author Response

“While the findings are novel, the authors should refer to STROBE guidelines for reporting observational studies and restructure the manuscript accordingly. May not include all points but some revisions would enhance the quality of this article. The conclusions are otherwise consistent with the evidence presented.”

We are grateful for your positive response! We have restructured the manuscript according to STROBE. We have included the STROBE Statement checklist and marked relevant text from the manuscript where possible and relevant.”

Reviewer 2 Report

The authors conducted a retrospective analysis on more than 1800 patients who were in their third trimester of pregnancy with a group B streptococci (GBS) swap collected. They aimed to evaluate the presence of specific vaginal microbes which may have a positive or negative correlation with the presence of GBS. The data showed that the presence of any lactobacillus or L. crispatus inversely correlated with GBS recovery. Other lactobacilli, gardnerella vaginalis, mixed aerobic bacteria as a group, and yeasts were not significantly correlated with GBS detection. These results were in agreement with some previous human studies and in disagreement with some mouse experiments, which were well-discussed in the manuscript. Overall, this study provided valuable results and that will encourage future development of using lactobacilli probiotic or metabolites for GBS prevention during pregnancy. Some suggestions are below:

1.     Please revisit Table 1, as some percentages do not add up to exact 100%.

2.     In Table 2, The “Positive culture” and “Negative culture” under the title of “GBS colonization” is very misleading. The readers may take them as the positive or negative for GBS, but in fact they are the positive/negative for the microbes listed to the left with a positive GBS culture. Please clarify.

Author Response

“Overall, this study provided valuable results and that will encourage future development of using lactobacilli probiotic or metabolites for GBS prevention during pregnancy”

Many thanks for your comments. Regarding your suggestions, see our point-by-point replies below.

Suggestion #1:

Please revisit Table 1, as some percentages do not add up to exact 100%.”

Revised as requested. The problem was in the format of the "Other" category within the "Mixed aerobic bacteria" group. This has now been corrected and it adds up to exact 100%. Thank you for your careful revision!

Suggestion #2:

“In Table 2, The “Positive culture” and “Negative culture” under the title of “GBS colonization” is very misleading. The readers may take them as the positive or negative for GBS, but in fact they are the positive/negative for the microbes listed to the left with a positive GBS culture. Please clarify.”

Revised as requested. Hopefully this is now clearer for the readers. We have corrected the title of the columns. We feel that the proportions in column 3 and 4 give the readers an impression of the magnitude of correlation and would suggest to leave it in the table, although it is in a way redundant. Thank you for your suggestion!